# New infant formulas for healthy term infants: A randomized, controlled, double-blind, multicenter, non-inferiority design safety study

Stephen A. Fleming[1], Stefanie Flunkert[2], Anne S. Kvistgaard[3], James McGrath[4], David K. Glover[5]*

1 Traverse Science, Mundelein, Illinois, United States of America, 2 BioDoks e.U., Georgen an der Stiefing, Austria, 3 Arla Foods Ingredients Group P/S, Viby J, Denmark, 4 Building Block Nutritionals, LLC, Charlottesville, Virginia, United States of America, 5 PBM Capital Group, LLC, Charlottesville, Virginia, United States of America

* dglover@pbmcap.com

## Abstract

### Objectives

Two infant formulas with unique combinations of 1,3-dioleoyl-2-palmitoyl-sn-glycerol sn2 palmitate, galactooligosaccharide, polydextrose, fructooligosaccharide, β-carotene, lutein, α-lactalbumin, osteopontin, and lactoferrin were evaluated for non-inferiority compared to a commercially available formula.

### Methods

In a randomized, controlled, parallel-arm, double-blind, multicenter, non-inferiority study, eligible infants were enrolled to receive an experimental (BBN-001 [Part 1; N = 129], BBN-102 [Part 2; N = 117]) formula or commercial formula (Brand; N = 143) for 120 days (Clinical Trials.gov NCT03331276). Infants were considered eligible if they were healthy, term (≥ 37 and ≤ 42 weeks of gestation), singleton newborns, with a birth weight of at least 2,500 g, and no more than 14 postnatal days-of-age. Anthropometric growth, formula intake, gastrointestinal tolerance, and adverse events were measured throughout the study, and fecal soap fatty acids were measured at the end. The primary endpoint was weight gain at the end of the trial, with treatment groups to be considered non-inferior if their weight gain was > −3 g/d compared to the control group.

### Results

Both experimental formulas were non-inferior to the Brand formula according to anthropometric outcomes. Formula intake, total adverse events, and stool frequency and consistency were similar to Brand formulas. Some measures of gassiness and fussiness improved in the experimental formulas (P < 0.05). Fecal calcium increased

**Data availability statement:** Supplementary Methods and Data are available in the files provided. Supplementary data contain summary data, including those used for visualization. Raw data may be made available upon request to corresponding author David Glover (dglover@pbmcap.com) or Cynthia Barber (study sponsor, cynthia.barber@munchkin.com, Vice President of Scientific and Clinical Affairs at Munchkin).

**Funding:** Building Block Nutritionals, LLC, USA (BBN) funded and conducted the clinical study, and BBN, Arla Foods Ingredients Group P/S, Denmark (https://www.arlafoodsingredients.com), and Munchkin, Inc., USA (https://www.munchkin.com) funded the writing of this manuscript. All three companies were involved in the decision to publish and the preparation of this manuscript.

**Competing interests:** I have read the journal's policy and the authors of this manuscript have the following competing interests: ASKV is an employee of Arla Foods Ingredients Group P/S, Denmark. SAF has ownership in and employment with Traverse Science. Arla Foods Ingredients Group P/S contracted Traverse Science to write the manuscript. SF has ownership of BioDoks e.U. and was contracted by Traverse Science to assist with manuscript preparation. JM is an employee of Building Block Nutritionals, LLC, USA. DG is an employee of PBM Capital Group, LLC, USA.

and fecal palmitic acid soaps decreased in both experimental formulas (all $P \leq 0.045$), and total soap fatty acids were decreased in the BBN-102 group compared to the Brand group ($P = 0.020$).

## Conclusions

The experimental formulas were well tolerated and deemed non-inferior to those of a Brand formula. The experimental formulas improved some measures of gastrointestinal tolerance compared to standard commercially available infant formulas.

## Trial registration

ClinicalTrials.gov NCT03331276

---

## 1. Introduction

It is widely accepted that breastfeeding is optimal and highly recommended over formula feeding [1], offering greater protection from infection [2] and support for cognitive development [3,4]. When breastfeeding is not an option, infant formula represents the sole source of nutrition. This clinical research study aimed to assess the non-inferiority of a new formula containing a novel combination of ingredients compared to a commercially available formula. Here, we assessed the following: 1,3-dioleoyl-2-palmitoyl-sn-glycerol sn2 palmitate (OPO), galactooligosaccharide (GOS), polydextrose (PDX), fructooligosaccharide (FOS), β-carotene, lutein, α-lactalbumin, osteopontin (OPN), and lactoferrin.

OPO, a triglyceride with 70–75% of the palmitic acid attached to the middle position of the glycerol backbone, is highly concentrated in human milk compared to bovine milk and vegetable oils [5,6]. Infants fed formula containing OPO have been shown to have reduced excretion of insoluble soaps formed from binding divalent cations, resulting in improved fatty acid and mineral absorption [5]. α-lactalbumin, lactoferrin, and osteopontin are highly abundant proteins in human milk compared to bovine milk [7–11]. α-lactalbumin is highly bioavailable, rich in tryptophan, and has been shown to support growth in infant formula [7–9]. Lactoferrin promotes iron-binding [12], immunity [11], and brain development [13], and its consumption in formula is well-tolerated [14]. Osteopontin, enriched in colostrum compared to term milk [11], is a multifunctional and highly phosphorylated glycoprotein, whose inclusion in formula supported growth while bringing the rates of fever and cytokine levels of formula-fed infants closer to that of breastfed infants [15]. Oligosaccharides are the third most concentrated solid in breast milk, with functions related to gastrointestinal, immune, and brain development [16]. While not considered human milk oligosaccharides, infant formulas containing GOS, PDX, and FOS have been widely studied and used in commercial formulas [17–19]. Lastly, carotenoids such as lutein and β-carotene may improve brain development [20] and are well tolerated in infant formula [21,22].

In this study, we tested the non-inferiority of two experimental formulas containing unique combinations of human milk components over 120 days in newborn infants in

a randomized, controlled, parallel-arm, double-blind, multicenter clinical study. We specifically investigated novel combinations of OPO, GOS, PDX, FOS, β-carotene, lutein, α-lactalbumin, OPN, and lactoferrin. The primary objective of this study was to assess whether the experimental formulas were non-inferior to a commercially available (Brand) formula in terms of growth and to compare the safety and tolerability of the experimental formulas with the Brand formula.

## 2. Materials and methods

### 2.1 Ethics

The study was conducted in accordance with Good Clinical Practice (GCP) as required by the International Council on Harmonisation of Technical Requirements for Registration of Pharmaceuticals for Human use (ICH) guidelines and in accordance with country-specific laws and regulations governing clinical studies of investigational products. Compliance with these requirements also constitutes conformity with the United States (US) 21 Code of Federal Regulations (CFR) Part 50 (Protection of Human Subjects), US 21 CFR Part 56 (Institutional Review Boards [IRB]), ICH Tripartite Guideline, Guideline for GCP E6 (R1), the Nuremberg Code, and the ethical principles of the Declaration of Helsinki. The IRB reviewed and approved this study's protocol and the Informed Consent Form (ICF). All parents or guardians needed to participate in the informed consent process and sign the ICF before any protocol-required procedures were performed.

### 2.2 Study design

This randomized, controlled, double-blind, parallel-arm, multicenter, non-inferiority study of healthy term formula-fed infants was designed following Good Laboratory Practice guidelines and the requirements for quality factors for infant formulas described in US 21 CFR 106.96. This study was conducted in two parts. The first (Part 1) was performed at 20 clinical sites in the United States between September 11th, 2017 and January 7th, 2019 (NCT03331276 registered on ClinicalTrials.gov on November 2nd, 2017 at https://www.clinicaltrials.gov/study/NCT03331276). The second (Part 2) included a subset of infants from Part 1 (Brand formula group) and enrollment (starting October 15th, 2020) of additional infants across 15 sites, for a total of 35 sites, with the trial ending on November 9th, 2021. Parts 1 and 2 followed the same protocol, except where otherwise noted. The original trial protocol and latest version are available in the Supplemental Material in S1 File. There were no deviations from the final trial protocol, however the final protocol differed from the first approved version in the following characteristics: 1) revision of the design from an equivalence to non-inferiority design (to confirm the experimental formulas did not lead to worse outcomes than commercially available formulas), 2) addition of Part 2 (to assess a formula without osteopontin, which has not been approved in the US), and 3) addition of formula intake as an endpoint (to address if growth outcomes were related to intake). Quality assurance across sites was conducted by monitors as described in the Supplemental Material in S1 File and through adherence to the standard operating procedures described in the below sections.

**2.2.1 Eligibility and exclusion criteria.** To be eligible for this study, infants needed to be at the time of birth a healthy, term (≥ 37 and ≤ 42 weeks of gestation), and singleton newborn with a birth weight of at least 2,500 g. At the time of baseline visit, infants needed to be designated by a physician to be healthy and at most 14 postnatal days-of-age. Weight-for-age, length-for-age, head circumference-for-age, and weight-for-length needed to be within the range of ≥ 5th and ≤ 95th percentile according to the World Health Organization (WHO) growth charts for infants and children 0–2 years-of-age. Furthermore, infants needed to have exclusively consumed and tolerated cow's milk infant formula at enrollment but could have been breastfed before enrollment. Parents or legal guardians additionally needed to agree to feed the study formula as the sole source of nutrition for the duration of the study and to have read and voluntarily signed an ICF.

Exclusion criteria included anatomical and physiological defects of the respiratory tract, other congenital defects, evidence of chronic hepatic, gastrointestinal, renal, cardiac, pulmonary, or neurological diseases, a family history of cow's milk protein intolerance or allergy, and non-singleton births. Further exclusion criteria included a maternal history

with known adverse effects on the fetus or the newborn infant, such as active tuberculosis, perinatal infection, substance abuse, or diabetes. Gestational diabetes was acceptable if the infant's birth weight was < 4,300 g. Subjects were considered non-compliant if they consumed a single non-study formula feeding more than 10 times during the duration of 16-week study and/or consumed more than 3 complete days of non-study formula, defined as greater than 50% of the number of feedings in a 24-hour period.

**2.2.2 Recruitment, enrollment, randomization, and blinding.** The study was performed double-blinded as treatment assignments were concealed from care providers, site personnel, investigators/outcome assessors, statisticians, and parents/guardians. Parents or guardians of healthy term infants that fulfilled the inclusion criteria were approached for potential enrollment. After completion of the ICF, infants were randomly assigned to Brand or to one of the experimental treatment groups in a 1:1 ratio using an interactive web-based randomization system (IWRS), blocked by the formula group and stratified by infant sex to allow for an equal number of males and females in each formula group. Formulas were provided in powder form in composite cans of identical size and shape. Cans were labeled with a unique case number to mask identity. All personnel at the study sites and parents or guardians were unaware of the product identity or group allocations. The IWRS system was the only place where the study subject's randomization numbers were paired with formula case numbers to ensure blinding. An unblinded Medical Monitor was assigned to keep a copy of the randomization list and assignments in case of emergency should unblinding outside of the IWRS system become necessary.

## 2.3 Study schedule

Infants were to consume the formula for 16 weeks with 9 study visits over 120 days, including six in-person and three telephone follow-up visits. Anthropometric measures (weight, length, and head circumference), the Infant Characteristics Questionnaire (ICQ), and a 3-day Formula/Diet Record of intake volume were collected at the six in-person study visits on days 0, 15, 30, 60, 90, and 120. Site personnel interviewed the parent or guardian at each study visit to obtain information about study formula tolerance, stool characteristics, adverse events (AE), and all concomitant therapies or medications performed or used since the last study visit. During Visit 7 on day 90, parents or guardians were provided with a stool collection kit to collect stools within the last three days of the study. The kit was returned on the last visit on day 120. At the final visit, blood was collected from each subject by heel stick to measure serum cytokines. Three days after the first visit, all parents or guardians were called to check compliance with study feeding and inquire about the subject's well-being. For Visits 4, 6, and 8 on days 45, 75, and 105, subjects did not visit the study site, but parents or guardians were interviewed during a phone call (Table 1).

## 2.4 Feeding regimen and study formulas

The control groups were provided a Brand formula (Enfamil Premium® or Enfamil Infant®, Mead Johnson Nutrition, IN, USA). Enfamil Premium® was phased out in 2018. Eleven infants in Part 2 in the Brand group were provided Enfamil Infant® instead, which has identical macro- and micro-nutrients with slightly less linoleic acid, no nucleotides, and less ARA than Enfamil Premium®. The experimental formulas were designated as BBN-001 (Part 1) and BBN-102 (Part 2). All formulas were isocaloric (20 kcal/oz) cow's milk-based infant formulas for term infants. Polydextrose, GOS, arachidonic acid (ARA), and docosahexanoic acid (DHA) concentrations were lower in the BBN-001 formula than in the Brand formula. Conversely, the BBN-001 formula was supplemented with greater OPN, OPO, α-lactalbumin, lutein, β-carotene, lactoferrin, and FOS. The experimental formulas BBN-001 and BBN-102 were primarily different in their concentrations of protein, GOS, PDX, FOS, iodine, lutein, OPN, and lactoferrin. The nutrient composition for components that are different between all groups are listed in Table 2, and the full specifications are listed in the Supplemental Data in S1 Data.

Nutrient and stability testing was performed to ensure that formulas met quality requirements. Formulas were labeled for clinical use only and kept dry, protected from light, and at a temperature of 16–29°C. Parents or guardians were

**Table 1. Study Schedule.**

| Study visit | 1 | 2 | 3 | 4 | 5 | 6 | 7 | 8 | 9 |
|---|---|---|---|---|---|---|---|---|---|
| Study day (± 3 days) | 0 | 15 | 30 | 45 | 60 | 75 | 90 | 105 | 120 |
| Infant age(± 4 days)* | 9 | 24 | 39 | 54 | 69 | 84 | 99 | 114 | 129 |
| Screening Procedures[1] | X | | | | | | | | |
| Stool Characteristics and Tolerance Questionnaires | X | X | X | X | X | X | X | X | X |
| Concomitant Medications | X | X | X | X | X | X | X | X | X |
| Adverse Events | X | X | X | X | X | X | X | X | X |
| Dispense and Collect Study Formula | X | X | X | | X | | X | | X |
| Anthropometry[2] | X | X | X | | X | | X | | X |
| Medical History | X | X | X | | X | | X | | X |
| Infant Characteristics Questionnaire (ICQ) | X | X | X | | X | | X | | X |
| 3-day Formula/Diet Record[3] | X | X | X | | X | | X | | X |
| Telephone Contact[4] | X | | | X | | X | | X | |
| Dispense Stool Collection Kit | | | | | | | X | | |
| Blood and Stool Collection | | | | | | | | | X |

*Actual age of infants in each group at enrollment. Here, infant age represents 9 days plus the study day.

[1]Included Informed consent; Inclusion/Exclusion Criteria; Demography; Randomization; Infant Feeding History, Physical Exam.

[2]Included assessment of body weight, body length, and head circumference.

[3]Initial 3-day Formula/Diet Record was recorded for 3 days following the first study visit.

[4]Initial telephone contact was performed 3 days after enrollment.

Shaded columns indicate visits that were performed at the study site while white columns indicate visits performed by phone interview.

provided with instructions to mix the formula to 20 kcal/oz, to handle formula with clean hands, to clean cans prior to opening, not to freeze or microwave formula, to store already prepared formula in the refrigerator for no more than 24 hours, to use a bowl of warm water to warm formula, to feed formula at room temperature, and to return all unopened cans of formula at the final visit. Infants consumed Brand or experimental formulas *ad libitum* for 16 weeks. Compliance with the study feeding regimen was monitored approximately every two weeks. A standard set of interview questions was used at each clinic visit and during telephone follow-ups between clinic visits to ask about the consumption of study formula and other consumed foods, including other formulas.

## 2.5 Anthropometric measurements

Weight, length, and head circumference were measured using standard methods across all sites according to an established standard operating procedure adapted from the US Department of Health and Human Services Growth Charts Training [24]. Infants were weighed unclothed to the nearest 10 g using a calibrated scale, with repeated measurements taken until two values were within 10 g. Length was measured to the nearest 0.1 cm in the recumbent position using a calibrated board with a fixed headpiece and moveable footpiece, with two measurements required to be within 0.5 cm. Infants were measured without shoes or hair ornaments, wearing light underclothing or diaper. Head circumference was measured to the nearest 0.1 cm using a flexible, non-stretchable tape placed around the maximal head circumference, with two measurements required to be within 0.2 cm.

## 2.6 Sample collection and analysis

Covance CLS (IN, USA) and ARUP Laboratories (UT, USA) were used for all laboratory determinations unless a special or emergency test was required. Both laboratories adhered to Good Clinical Practices and provided all necessary

**Table 2. Nutrient composition of control and experimental formulas[1].**

| Component, per liter | Enfamil Premium[2] | BBN-001 | BBN-102 |
|---|---|---|---|
| Protein (g) | 14.1 | 15.5 | 17.6 |
| Fat (g) | 37.3 | 38.0 | 35.9 |
| OPO (g) | 0 | 4.9 | 4.9 |
| Carbohydrate (g) | 79.5 | 75.3 | 73.9 |
| Galactooligosaccharide (mg) | 2000.0 | 207.7 | 2111.7 |
| Polydextrose (mg) | 2000.0 | 211.2 | 0 |
| Fructooligosaccharide (mg) | 0 | 497.7 | 211.2 |
| Linoleic Acid (mg) | 5631.2 | 5983.2 | 5983.2 |
| Pantothenic Acid (µg) | 3519.5 | 4223.4 | 4223.4 |
| Iodine (µg) | 105.6 | 70.4 | 105.6 |
| β-carotene (µg) | 0 | 133.7 | 133.7 |
| Lutein (µg) | 0 | 126.7 | 70.4 |
| Nucleotides (mg) | 29.6 | 27.5 | 28.2 |
| L-Carnitine (mg) | ** | 9.9 | 9.2 |
| Taurine (mg) | ** | 40.8 | 42.2 |
| Osteopontin (mg)[3] | * | 126.7 | * |
| Lactoferrin (mg)[4] | * | 54.9 | 63.4 |
| Docosahexaenoic acid (mg) | 119.7 | 68.3 | 67.6 |
| Arachidonic acid (mg) | 239.3 | 136.6 | 133.7 |
| α-lactalbumin (g) | ** | 2.561 | 2.561 |

Abbreviations: g, gram; IU, international unit; µg, microgram; mg, milligram; OPO, 1,3-dioleoyl-2-palmitoylglycerol; NC, not calculable.

[1]Only components that differ between groups are shown. All formulas had equivalent amounts of: energy, vitamins A, C, D, E, K, B1, B2, B3, B6, and B12, folic acid, biotin, choline, inositol, calcium, phosphorus, magnesium, iron, zinc, manganese, copper, sodium, potassium, chloride, and selenium. Full specifications are available in the Supplemental Data in S1 Data.

[2]Data for Enfamil Premium® received from personal communication by email with company representative. Enfamil Premium® was phased out in 2018.

[3]The estimated average OPN content in five formulas reported by Schack et al. 2009 [23] is 8.6 mg/L.

[4]The content in infant formulas estimated by Chatterton et al. 2013 [11] is 48.1 mg/L.

*Not added to formula, though some amount may be present.

**Nutritional information not available.

supplies and shipping materials. Stool samples were collected three consecutive days before Visit 9 and in separate vials. To assess stool concentrations of total and palmitic soap fatty acids and total calcium, stool samples were collected on three consecutive days before Visit 9 in separate vials. The diaper was laid out flat for stool collection, and feces were collected with a provided scoop that was then placed into the specimen vial. The vial was firmly closed, sealed in the provided plastic hazard bag, immediately frozen at −20°C, and shipped frozen to Covance CLS (IN, USA) for analysis. Fecal calcium was analyzed by modifying AOAC 984.27, 985.01, and 2011.14 on a dry weight basis. Soap fatty acids were determined via gas chromatography using methods previously described [25], and reported on a dry weight basis. See the Supplemental Material in S1 File for a more detailed description of analyses, including collection of blood and analysis of cytokines.

## 2.7 Adverse events and gastrointestinal tolerance

Adverse events were monitored, by parent or guardian interview at each site visit, and recorded throughout the study and rated according to severity and relation to formula. Standardized definitions for common gastrointestinal (GI) symptoms,

such as stool issues, spit-up, vomiting, gastroesophageal reflux disease, and issues with crying or skin were agreed on by the investigators and medical monitor used for reporting AE, using the Medical Dictionary for Regulatory Activities (MedDRA) versions 20.0 and 23.0 for Parts 1 and 2, respectively. A serious adverse event was defined as an AE that results in death, is life-threatening, causes significant, persistent, or permanent changes, impairments, damage, or disruptions in body function or structure, physical activities, or quality of life; requires or prolongs hospitalization; or is otherwise medically significant and, based on appropriate medical judgment, may jeopardize the subject and require medical or surgical intervention.

The parent or guardian filled out the ICQ survey [26], which rates parent report of infant behavior as fussy, dull, unpredictable, and unadaptable. Parent report of stool frequency (number of movements in the previous 24-hr period), consistency (hard = 1, formed soft = 2, mushy soft = 3, runny soft = 4, watery = 5), and gassiness/fussiness (less, about the same, or more than normal) were used to assess gastrointestinal tolerance.

## 2.8 Planned statistical analyses

All analyses were performed using SAS version 9.4 or higher (SAS Institute Inc., NC, USA) according to an *a priori* designed statistical analysis plan, described henceforth. Parts 1 and 2 were analyzed as distinct trials given they were conducted at different times on different populations. Further, the objective of the study was to assess non-inferiority of the experimental formulas to the "standard-of-care", a commercially available formula. Thus, comparisons between the BBN-001 and BBN-102 formulas were not made. For analysis of growth-related outcomes (weight, length, and head circumference gain velocities) between experimental and Brand formulas, a mixed-effect model with repeated measures (MMRM) with gender, age, baseline value, formula, visit, site, formula by-site interaction, and formula by-visit interaction as factors was used. The covariance matrix was chosen based on the Akaike information criteria (AIC) to choose the covariance matrix that best fit the model. Non-inferiority was achieved if the lower limit of the two-sided 95% CI for the formula difference was greater than the non-inferiority margin on Visit 9, see the following statistical hypothesis.

$$H_0: \ T - C \ \leq \ -m \ (\text{inferiority})$$

$$H_1: T - C \ > \ -m \ (\text{non} - \text{inferiority})$$

A margin of 3 g/day for weight gain at Visit 9, the primary endpoint, has been used in previous non-inferiority studies [27,28] and has been recommended by the Institute of Medicine as a clinically significant threshold for assessing safety [29]. No published recommendations were identified for the non-inferiority margin for length and head circumference gain velocities, nor anthropometric z-scores. After personal communication with the FDA, margins were selected at 20% of the Brand's value at Visit 9 for length and head circumference gain, and Z-score anthropometrics at 0.84. Comparison of *P*-values to an α threshold was not used as part of the assessment for any growth-related outcomes. Secondary endpoints (length and head circumference gain, anthropometric z-scores, and formula intake) were tested for non-significance using the Hochberg step-up multiple testing procedure at α = 0.05 and Hochberg-adjusted confidence intervals. Differences between groups per visit for outcomes not based on non-inferiority (visits other than Visit 9 and stool composition outcomes) were assessed via a two-sample t-test. All statistical tests were considered significant at α ≤ 0.05, unless noted otherwise. Frequency of AE was analyzed using Fisher's exact test. Bowel movements in the past 24 hours, stool consistency, and ICQ scales were modelled using mixed-model for repeated measures (MMRM) with gender, age, baseline value, formula, site, visit, and formula by visit interaction as factors. Gassiness and fussiness (counts of less, about the same, or more than normal) were analyzed via generalized linear model with repeated measures with sex, age at

enrollment, baseline value, formula, visit, and site. Only the difference at visit 9 was assessed for bowel movements, stool consistency, ICQ scales, gassiness, and fussiness. Summary data (descriptive statistics and results of analyses) are available in the Supplemental Data in S1 Data.

**2.8.1 Subject populations.** The Intent-to-Treat population (ITT) included all subjects enrolled and randomized to one of the study groups. The Safety population included all randomized subjects that consumed the assigned formula at least once. The Safety population was used for analysis of AE, stool composition, and parent-report of gastrointestinal tolerance and ICQ data. The Per Protocol population (PP) included a subset of the ITT, including all subjects that completed the feeding protocol without major protocol deviations or that consumed a non-study formula no more than ten times during the study. Infants that consumed non-study formulas for over three full days, defined as greater than 50% of the number of feedings in 24 hours, were also excluded from the PP population. Given some sites relied on paper growth charts and small errors could occur with subjects slightly outside the 5th and 95th percentile for anthropometric scores, if one of four anthropometric inclusion criteria were not met, the infant was retained in the PP population. However, subjects were excluded from the PP if inclusion criteria were not met by two or more of the four anthropometric measurements (weight-for-age, length-for-age, head circumference-for-age, and weight-for-length). Since this study aimed to demonstrate non-inferiority, the PP was designated to represent the primary analysis population to evaluate the treatment groups in terms of efficacy.

Subjects that enrolled in the study but never received the assigned formula were dropouts. Dropouts were not replaced. Subjects that did not complete the full study period due to protocol adherence or who requested to be discontinued were not replaced. Analyses were carried out with the available data without imputation for missing data. Analyses of ITT data using mixed models without imputation has been shown to have greater statistical power than alternative methods using imputation such as last-observation carried forward, best-value replacement, and worse-value replacement [30], hence why it's possible to conduct an ITT analysis with a smaller sample size than the number of enrolled subjects.

**2.8.2 Sample size and statistical power.** For Part 1, sample size estimation was based on the primary endpoint weight gain velocity (g/d) over a 16-week study period. Assuming a standard deviation in weight gain of 5.6 g/d [31] and 80% power, 90 subjects per formula group (180 total) was estimated to be sufficient to demonstrate non-inferiority with a margin of 3 g/d. An interim analysis, completed blind when 128 subjects (50% of the total goal sample size) were enrolled and completed the 2-month time point, revealed the actual standard deviation to be 6.0 g/d. The power analysis was updated to reflect 90 subjects per group (180 subjects total) were necessary to achieve 90% power. Assuming a 25% attrition rate, 256 subjects of equal sex were to be enrolled in a 1:1 allocation ratio to the control and treatment groups.

For Part 2, in addition to the infants randomized to the Brand formula in Part 1, additional infants were randomized to the Brand formula or BBN-102 groups in a 1:8 ratio. Assuming a 25% attrition rate, standard deviation in weight gain of 6.0 g/d, and to obtain 90% power, 10 (totaling 139) and 86 additional subjects were estimated to be randomized to the Brand and BBN-102 groups, respectively.

**2.8.3 Ancillary statistical analyses.** The following analyses were not specified *a priori* but conducted to aid in the interpretability of results: conversion of concomitant medication use, medical histories, and frequency of adverse events into relative risks (see Supplemental Data in S1 Data), analysis of concomitant medication and medical history by Fisher's Exact test, anthropometrics were contextualized by overlaying them with the WHO growth curves (https://www.who.int/tools/child-growth-standards/standards), the actual age of infants was estimated by adding the average age at enrollment for each group to the study day, and between-group differences (with no inferiority margin) were measured via two-sample t-test for all anthropometric outcomes. Analyses for cytokines were planned, however in both Parts 1 and 2 the median value of nearly all cytokines was below the detectable limit, and numerous samples did not have enough volume for laboratory analyses. Given the lack of analyzable data, no comparisons of cytokines were made between groups.

 

## 3. Results

### 3.1 Subject disposition and demographics

In total 389 infants were randomized, infants randomized to the Brand formula in Part 2 represented 129 infants from Part 1 and 14 additional infants randomized to the Brand formula in Part 2 (Fig 1). Approximately 67.5–76% of subjects completed the trial, with most dropout related to infants that were lost to follow-up (5.4–10.5% of enrolled) or withdrawn by parents/guardians (10.9–18.6% of enrolled). The Per-Protocol population included only infants that completed the study protocol without major protocol deviations. Loss between the PP and ITT populations in the Brand group was largely related to infants not receiving study formulas for over 3 full days. Otherwise, all groups exhibited similar rates of deviations related to infants enrolled despite not meeting inclusion/exclusion criteria. Two subjects in the BBN-102 group were excluded from the PP population given exposure to COVID-19 and missing data.

Subject characteristics at enrollment between Brand and BBN-001 groups were similar across all measures in Part 1 (Table 3). In Part 2 compared to the Brand group, the BBN-102 group was comprised of a statistically greater proportion of infants reported as black or African American and Hispanic or Latino, a lesser proportion of infants exposed to smoke at home, and on average shorter body lengths and head circumferences at enrollment (all $P < 0.05$). However, sex, age, birthweight, enrollment body weight, and rates of breastfeeding and maternal smoking history were similar between Brand and BBN-102 groups.

In Part 1, there was no difference in concomitant medications between the Brand and BBN-001 groups (all $P \geq 0.059$, Supplemental Data in S1 Data). In Part 2, the BBN-102 group exhibited higher rates of using gripe water (5 [4.9%] vs 0 infants, $P = 0.018$) and lesser rates of using Ranitidine (0 vs 7 [5.6%] infants, $P = 0.017$). Uses of gripe water in the BBN-102 group were related to gassiness, fussiness, or hiccups. Two of seven uses of Ranitidine in the Brand group were related to preventative use or medical history, the remaining five were related to an AE. Medical histories were similar between groups and/or unrelated to the treatments (e.g., circumcision) (Supplemental Data in S1 Data).

### 3.2 Growth and formula intake

No differences in non-inferiority results between the ITT and PP populations were observed, thus all data herein describe the PP population. Across both Parts 1 and 2, for weight, length, and head circumference gain velocities, the lower 95% confidence interval exceeded the inferiority margin on Visit 9, and thus the BBN-001 and BBN-102 formulas were considered non-inferior to Brand formulas (Table 4). Growth plots compared to WHO 0–2 years old infants' growth curves further validated that the growth of all infants was in an acceptable range, with both the means and confidence intervals of all groups at all visits within 1 Z-score of zero (Figs 2 and 3). In Part 1, compared to the Brand group, the BBN-001 group exhibited a higher BW gain ($P < 0.05$, Table 4) and weight-for-length Z-score on Visit 9 ($P < 0.05$, Fig 2G), and had a higher head circumference-for-age Z-score at Visit 3 only ($P < 0.05$, Fig 2F). In Part 2, the BBN-102 group exhibited a higher head circumference in cm and Z-scores throughout the trial (all $P < 0.05$, Fig 3EF), but head circumference gain was similar at Visit 9 (Table 4). Body length was shorter in the BBN-102 group at enrollment ($P < 0.05$, Fig 3C), however length gain at Visit 9 was higher than the Brand group ($P < 0.05$, Table 4) despite similar length-for-age Z-scores throughout the trial ($P < 0.05$, Fig 3D). No significant differences in formula intake were found between the treatment and control groups at any time (all $P \geq 0.367$, Fig 4).

### 3.3 Adverse events

Adverse events are detailed in the safety population, which includes infants who consumed formula at least once, regardless of trial completion. In both Parts 1 and 2, over half of all infants experienced at least one AE during the study, of which ≥ 50% were classified as moderate or mild, and ≥ 35% deemed unrelated to the formula (Table 5). Less than 2.6% of infants in each group experienced a serious AE, and ≤ 6.9% per group had an AE event that led to formula

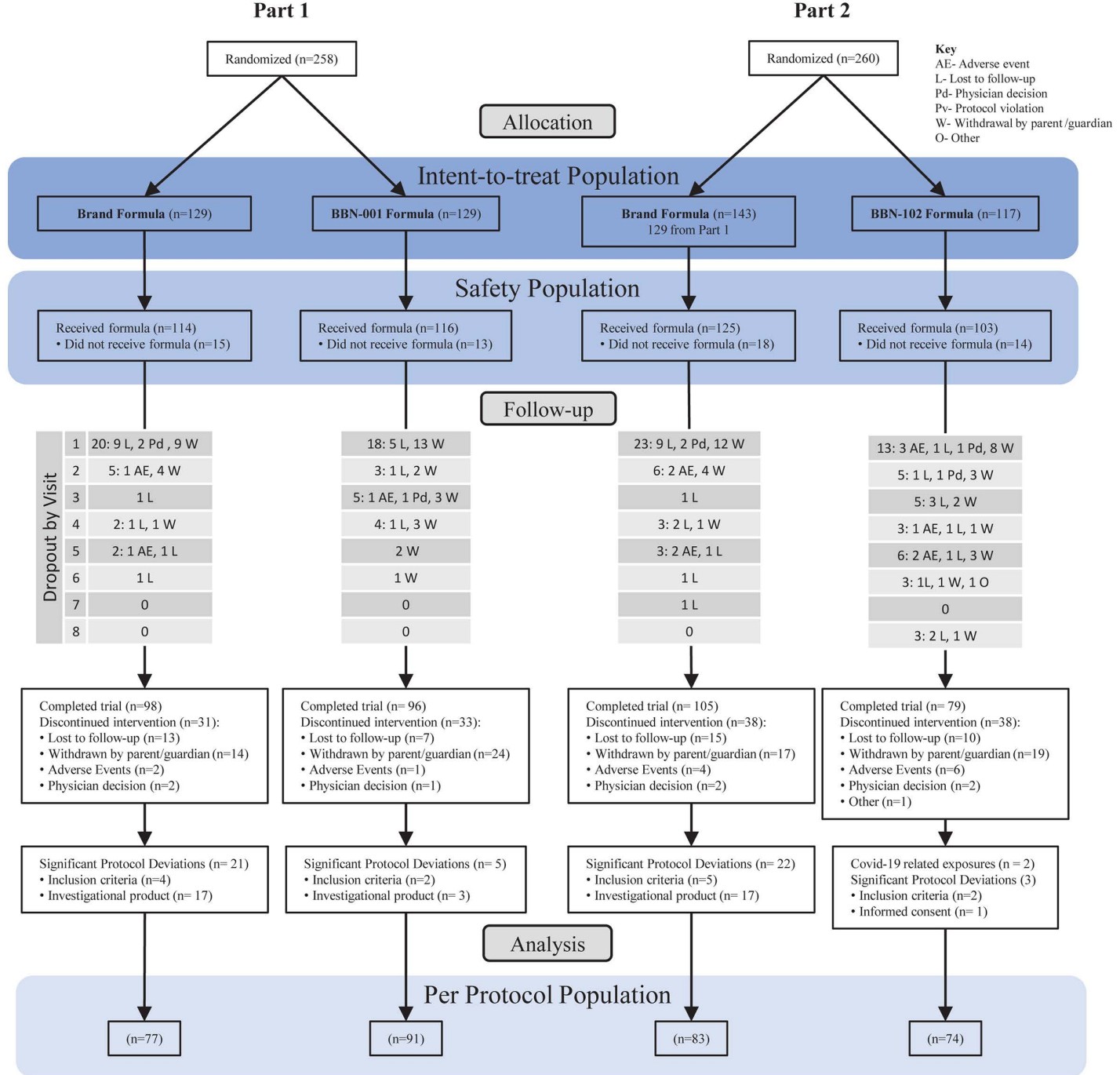

**Fig 1. Flow of subjects from allocation through analysis.**

discontinuation. Adverse events of particular interest (AESI) accounted for less than 27% of all reported AEs per group, with the majority of AESIs associated with gastrointestinal issues such as gastroesophageal reflux disease, vomiting, constipation, and spit-up.

**Table 3. Demographics of the Per Protocol population.**

| | Part 1 | | Part 2 | |
|---|---|---|---|---|
| **Criteria** | **Brand** | **BBN-001** | **Brand** | **BBN-102** |
| n | 77 | 91 | 83 | 74 |
| Female, n (%) | 38 (49.4) | 45 (49.5) | 42 (50.6) | 44 (59.5) |
| Race, n (%) | | | | |
| White | 68 (88.3) | 80 (87.9) | 73 (88.0) | 49 (66.2)* |
| Black or African American | 7 (9.1) | 10 (11.0) | 8 (9.6) | 25 (33.8)* |
| Asian | 0 | 0 | 0 | 1 (1.4) |
| Native Hawaiian or Pacific Islander | 0 | 0 | 0 | 1 (1.4) |
| American Indian or Alaskan Native | 0 | 1 (1.1) | 0 | 0 |
| Other | 2 (2.6) | 0 | 2 (2.4) | 0 |
| Ethnicity, n (%) | | | | |
| Non-Hispanic or Latino | 62 (80.5) | 71 (78.0) | 64 (77.1) | 52 (70.3) |
| Hispanic or Latino | 15 (19.5) | 20 (22.0) | 18 (21.7) | 21 (28.4) |
| Unknown or not reported | 0 | 0 | 1 (1.2) | 1 (1.4) |
| Infant anthropometrics, Mean±SD | | | | |
| Infant's birth weight (kg) | 3.37±0.412 | 3.29±0.375 | 3.38±0.419 | 3.30±0.388 |
| Enrollment age (days) | 8.8±4.25 | 9.4±4.22 | 8.8±4.3 | 7.9±3.62 |
| Enrollment body length (cm) | 50.59±1.796 | 50.55±1.843 | 50.6±1.834 | 49.91±1.615* |
| Enrollment body weight (kg) | 3.40±0.407 | 3.34±0.370 | 3.41±0.410 | 3.31±0.406 |
| Enrollment head circumference (cm) | 35.26±1.339 | 35.14±1.242 | 35.28±1.311 | 34.66±1.088* |
| Maternal Smoking History? Yes, n (%) | 4 (5.2) | 8 (8.8) | 5 (6.0) | 4 (5.4) |
| Of yes, number of cigarettes per day, Mean SD | 8±8.12 | 9±7.54 | 8.4±7.09 | 5.3±3.4 |
| Is the infant exposed to smoke at home? Yes, n (%) | 8 (10.4) | 13 (14.3) | 8 (9.6) | 0 (0.0)* |
| If yes, number of cigarettes per day, Mean SD | 14±12.17 | 13.5±7.85 | 14±12.17 | NA |
| Was infant ever breastfed? Yes, n (%) | 33 (42.9) | 39 (42.9) | 37 (44.6) | 30 (40.5) |
| Duration (days), Mean±SD | 4.7±3.7 | 5.1±3.97 | 4.8±3.66 | 5.6±3.63 |

Abbreviations: n, sample size; SD, standard deviation.

*$P<0.05$ compared to Brand group, analyzed by two-sample t-test for continuous data and Fischer's Exact test for categorical data.

In Part 1, fewer infants in the BBN-001 group exhibited dermatitis diaper compared to the Brand group (0% vs 6.1%, $P=0.007$), otherwise there was no difference in the frequency of AEs. In Part 2, the BBN-102 group exhibited fewer AEs of moderate severity or that were related to formula (9.7% vs 20.8%, $P=0.028$), however there was no specific type of formula-related AE that was different in frequency between groups.

### 3.4 Stool characteristics and gastrointestinal tolerance

Stool characteristics and gastrointestinal tolerance were assessed in the Safety population. In Part 1, the BBN-001 group exhibited higher fecal calcium ($P<0.05$), lower palmitic acid soaps ($P<0.05$), and similar total soap fatty acids as the Brand formula group (Fig 5ACE). In Part 2, the BBN-102 group exhibited higher calcium ($P<0.05$), lower palmitic acid soaps ($P<0.05$), and lower total soap fatty acids ($P<0.05$) than the Brand formula group (Fig 5BDF).

In Part 1, there was no difference between formulas on Visit 9 in bowel movement consistency ($P=0.680$) and frequency ($P=0.200$), or in the odds of being "more fussy than normal" ($P=0.462$). However, for Visit 9, the odds of the Brand formula being higher in "more gas than normal" from parent ratings were 2.93 [1.40, 6.15] times higher than that of the BBN-001 group. In Part 2, for Visit 9, the odds of the Brand formula being higher in "more fussy than normal" and

**Table 4. Non-inferiority determination for growth outcomes in the Per Protocol population.**

| Outcome | Mean ±95% CI | Mean ±95% CI | LS mean difference ± 95% CI from Brand | Margin[1] | Lower 95% CI> inferiority margin? |
|---|---|---|---|---|---|
| **Part 1, Visit 9** | **Brand** | **BBN-001** | | | |
| BW gain, g/day | 27.4 [26.12, 28.63] | 29.7 [28.38, 30.99] | 1.780* [0.004, 3.556] | −3 | Yes |
| Length gain, cm/day | 0.105 [0.101, 0.108] | 0.103 [0.100, 0.107] | −0.002 [-0.007, 0.003] | −0.020 | Yes |
| Head circumference gain, cm/day | 0.057 [0.054, 0.059] | 0.056 [0.054, 0.059] | −0.001 [-0.004, 0.002] | −0.012 | Yes |
| Weight-for-age Z-score | −0.20 [-0.388, -0.002] | 0.05 [-0.135, 0.240] | 0.303* [0.091, 0.515] | −0.84 | Yes |
| Length-for-age Z-score | −0.12 [-0.339, 0.105] | −0.23 [-0.435, -0.016] | −0.09 [-0.307, 0.120] | −0.84 | Yes |
| Head circumference-for-age Z-score | 0.63 [0.381, 0.885] | 0.49 [0.303, 0.682] | −0.048 [-0.291, 0.195] | −0.84 | Yes |
| Weight-for-Length Z-score | −0.09 [-0.304, 0.116] | 0.34 [0.090, 0.582] | 0.46* [0.154, 0.758] | −0.84 | Yes |
| **Part 2, Visit 9** | **Brand** | **BBN-102** | | | |
| BW gain, g/day | 27.8 [26.50, 29.01] | 28.9 [27.69, 30.09] | 1.27 [-0.580, 3.120] | −3 | Yes |
| Length gain, cm/day | 0.105 [0.102, 0.109] | 0.112 [0.108, 0.117] | 0.009* [0.004, 0.015] | −0.021 | Yes |
| Head circumference gain, cm/day | 0.057 [0.055, 0.060] | 0.057 [0.055, 0.059] | −0.001 [-0.005, 0.002] | −0.011 | Yes |
| Weight-for-age Z-score | −0.15 [-0.338, 0.031] | 0.01 [-0.172, 0.182] | 0.12 [-0.158, 0.405] | −0.84 | Yes |
| Length-for-age Z-score | −0.12 [−0.338, 0.099] | 0.09 [-0.123, 0.311] | 0.14 [-0.208, 0.479] | −0.84 | Yes |
| Head circumference-for-age Z-score | 0.66 [0.440, 0.887] | 0.34 [0.114, 0.569] | −0.43* [-0.773, -0.085] | −0.84 | Yes |
| Weight-for-Length Z-score | −0.03 [-0.244, 0.176] | −0.01 [-0.216, 0.207] | 0.13 [-0.186, 0.441] | −0.84 | Yes |

Abbreviations: LS, least squares; CI, confidence interval.

[1]Margin was chosen *a priori* as 3 g/d for weight gain velocity, 20% of the Brand formula's value at Visit 9 for length and head circumference gain velocities, and Z-scored anthropometrics at 0.84. Differences between groups were estimated from a mixed-effect model with repeated measures (MMRM), with gender, age, baseline value, formula, visit, site, formula by-site interaction, and formula by-visit interaction as factors.

*95% CI does not overlap zero, indicating a statistically significant difference between groups.

"more gas than normal" from parent ratings were 2.24 [1.11, 4.52] and 3.80 [1.85, 7.82] times that of Brand, respectively. BBN-102 was not different in bowel movement number or consistency from the Brand formula ($P \geq 0.513$).

Evaluation of ICQ data at Visit 9 showed no significant differences between the BBN-001 formula and Brand formula groups for the subscale components fussy/difficult ($P=0.211$), unpredictable ($P=0.524$), unadaptable ($P=0.976$), or dull ($P=0.415$). BBN-102 exhibited lower average ratings (−2.6 [−4.5, −0.8]) on the fussy/difficult subscale, but not on scales for unpredictable ($P=0.379$), unadaptable ($P=0.257$), or dull ($P=0.106$).

## 4. Discussion

In this randomized, controlled, parallel-arm, double-blind, multicenter, non-inferiority trial, healthy term infants were fed either an experimental or commercially available formula. Weight, length, and head circumference gain velocities of both

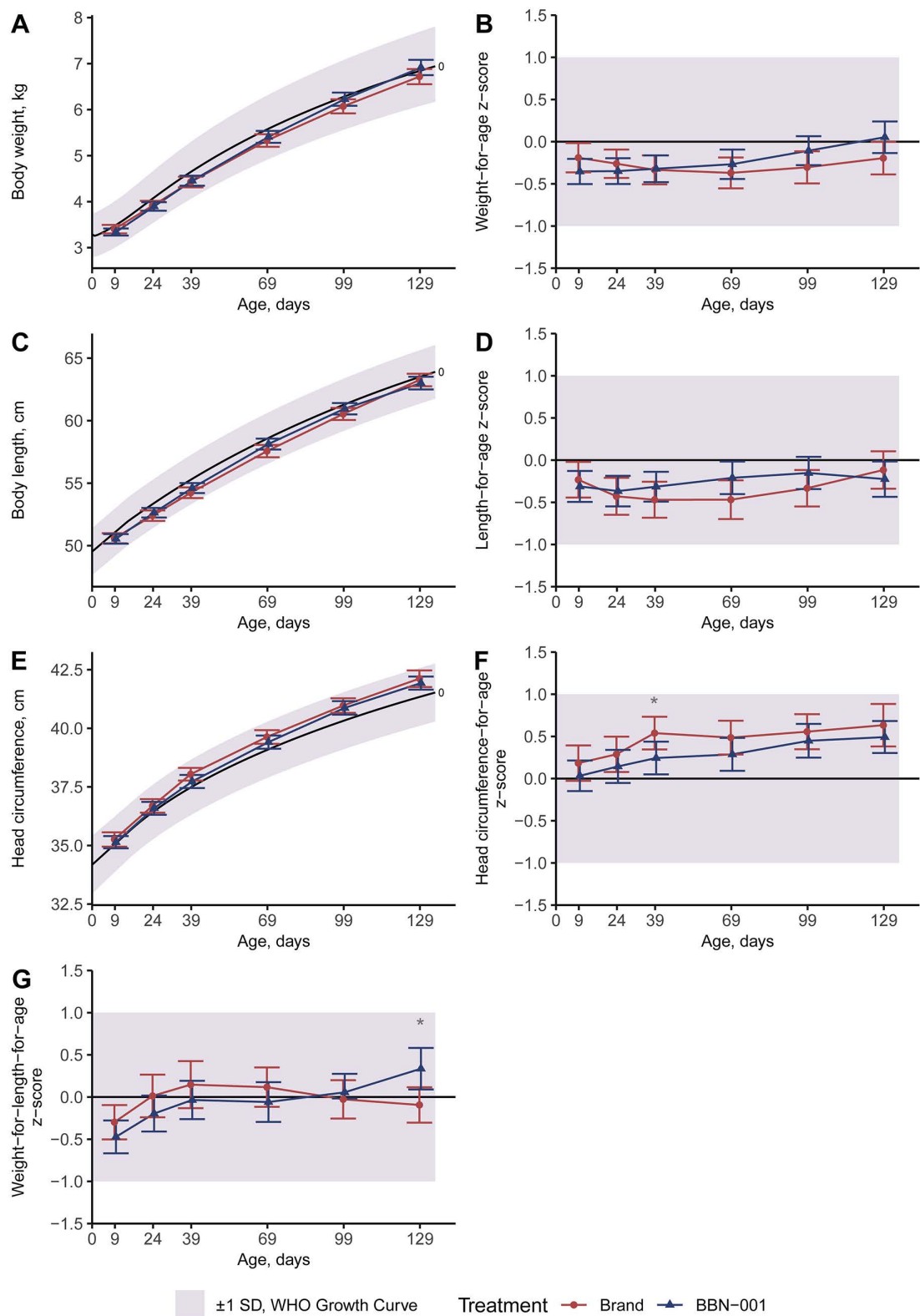

**Fig 2. Part 1 anthropometrics in the Per Protocol population.** All plots represent means and 95% confidence intervals. Ages are adjusted to reflect postnatal age, not day of study. The shaded region represents the WHO growth curves within 1 standard deviation (Z-score) for each measurement. *Two sample t-test indicates significance between groups at P<0.05.

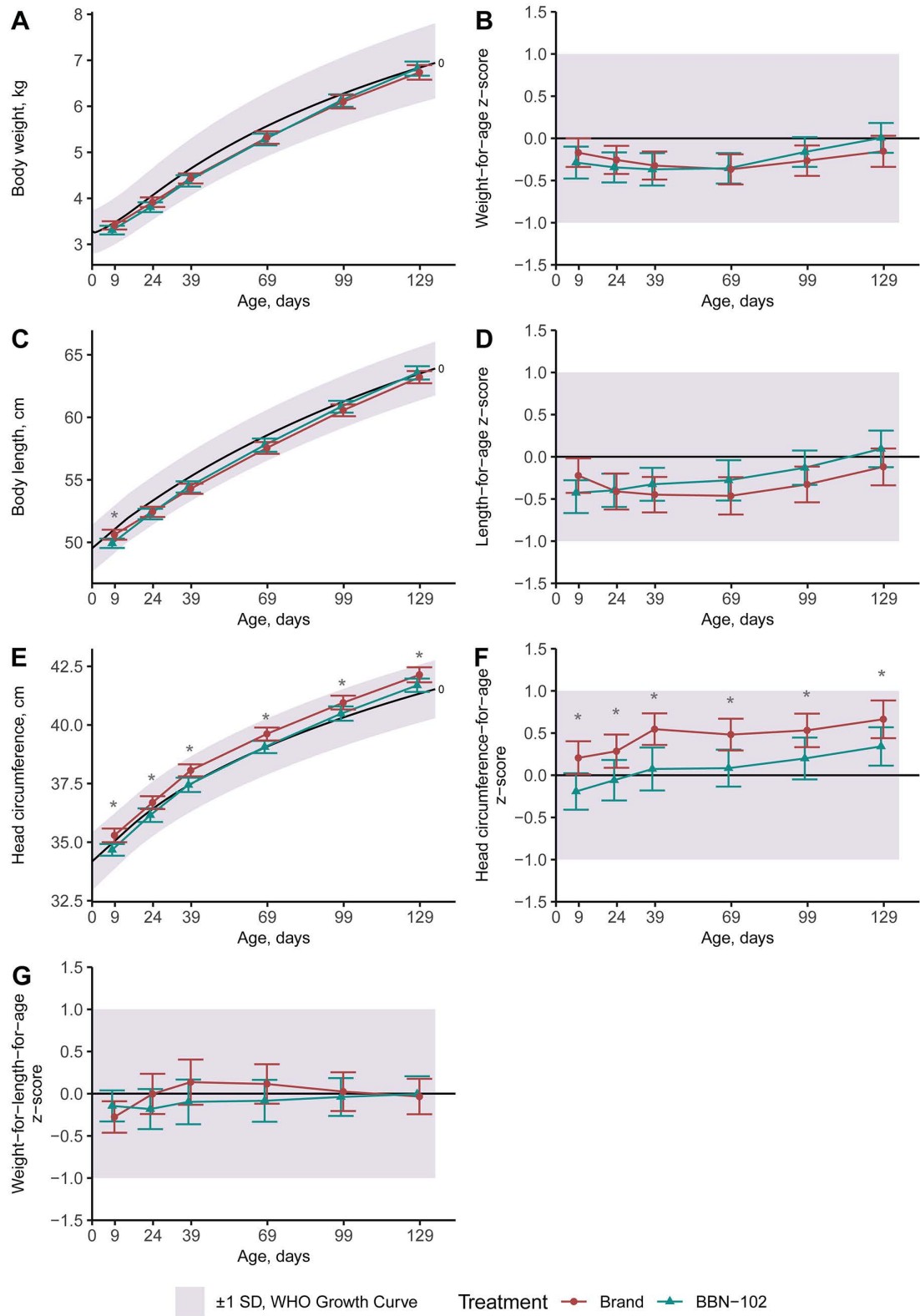

**Fig 3. Part 2 anthropometrics in the Per Protocol population.** All plots represent means and 95% confidence intervals. Ages are adjusted to reflect postnatal age, not day of study. The shaded region represents the WHO growth curves within 1 standard deviation (Z-score). *Two-sample t-test indicates significance between groups at P < 0.05.

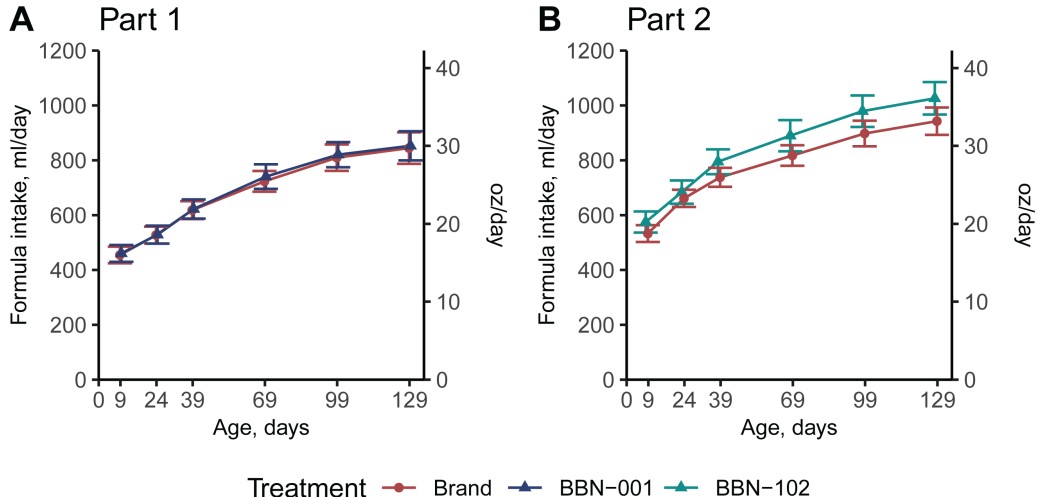

**Fig 4. Formula intake across duration of the study in the Per Protocol population.** Ages are adjusted to reflect postnatal age, not day of study. Two sample-test revealed no differences between groups at any time measured (all $P \geq 0.367$). Values represent the mean ± 95% confidence interval.

the BBN-001 and BBN-102 formulas were found non-inferior to the Brand formula, and there were no significant differences in formula intake compared to the Brand formula. Compared to Brand formulas, there were similar incidences of total adverse effects, similar stool frequency and consistency, reductions in fecal total soap fatty acids and palmitic acid, increases in fecal calcium, and some measures of gastrointestinal tolerance indicated an improvement with the experimental formulas.

The growth patterns of all groups aligned with WHO growth curves for infants aged 0–2 years. While non-inferior, the BBN-001 group exhibited faster BW gain and a higher weight-for-length Z-score at Visit 9 with a transient increase in head circumference at Visit 3. Conversely, differences in anthropometrics for the BBN-102 group appeared to be related to enrollment differences, as head circumference size and Z-scores remained elevated from enrollment to completion. Then, body length at enrollment was shorter for the BBN-102 group however total length gain was ultimately greater by end of the trial. However, it should be noted that the study was designed and powered as a non-inferiority study, rather than an equivalence test. Given the present study measured a combination of ingredients, it's not possible to attribute any change to one ingredient, however recent meta-analyses are informative.

Despite some differences between groups, experimental groups exhibited non-inferior growth similar to that described in previous trials. A meta-analysis revealed infants fed Sn-2-palmitate-enriched formulas demonstrated higher weight gain (0.81 [0.23, 1.39] g/d), similar length gain (0.02 [−0.01, 0.05] cm/w), and head circumference (−0.006 [−0.015, 0.004] cm/w) compared to standard formulas [32]. For comparison the present study reports similar differences between groups in weight gain (1.27–1.78 g/d), length gain (−0.014–0.063 cm/w), and head circumference gain (−0.007 cm/w). A meta-analysis of numerous types of prebiotics in formula found that on average prebiotics do not affect weight, length, or head circumference gain or z-scores [33] over a 3-month period. However, sub-group analyses revealed GOS+PDX reduced weight gain (standardized mean difference of −0.63 [−0.89, −0.38]), while GOS alone, FOS alone, and GOS+FOS had lesser-to-no impact on weight. None of these combinations affected length or head circumference gain [33]. No meta-analyses exist for the other unique ingredients in the present study, however such ingredients have been shown to have no or small impacts on infant growth [(α-lactalbumin [7,8]; bovine osteopontin [15]; lutein [21,22]; β-carotene [22]; lactoferrin [14,34]). Ultimately, these non-inferiority results support and expand upon previous research by demonstrating that combinations of these ingredients are not inferior to a commercially available formula.

**Table 5. Adverse Events in the Safety population.**

| Category[1] | Part 1 | | | Part 2 | | |
|---|---|---|---|---|---|---|
| | Brand n (%) | BBN-001 n (%) | P-value[2] | Brand n (%) | BBN-102 n (%) | P-value[2] |
| N, total | 114 | 116 | | 125 | 103 | |
| Subjects with ≥ one Adverse Events (AE) | 66 (57.9) | 78 (67.2) | 0.173 | 70 (56.0) | 57 (55.3) | 0.999 |
| Severity | | | | | | |
| Mild | 39 (34.2) | 47 (40.5) | 0.343 | 40 (32.0) | 46 (44.7) | 0.056 |
| Moderate | 24 (21.1) | 27 (23.3) | 0.752 | 26 (20.8) | 10 (9.7) | 0.028* |
| Severe | 3 (2.6) | 4 (3.4) | 0.999 | 4 (3.2) | 1 (1.0) | 0.381 |
| Relation to formula | | | | | | |
| Not related or unlikely | 42 (36.8) | 50 (43.1) | 0.349 | 44 (35.2) | 47 (45.6) | 0.135 |
| Related or possibly | 24 (21.1) | 28 (24.1) | 0.637 | 26 (20.8) | 10 (9.7) | 0.028* |
| Flatulence | 9 (7.9) | 13 (11.2) | 0.502 | 9 (7.2) | 2 (1.9) | 0.117 |
| Irritability | 7 (6.1) | 6 (5.2) | 0.783 | 6 (4.8) | 2 (1.9) | 0.299 |
| Gastroesophageal reflux disease | 4 (3.5) | 4 (3.4) | 0.999 | 5 (4.0) | 0 | 0.066 |
| Constipation | 2 (1.8) | 5 (4.3) | 0.446 | 2 (1.6) | 2 (1.9) | 0.999 |
| Vomiting | 4 (3.5) | 3 (2.6) | 0.721 | 5 (4.0) | 1 (1.0) | 0.226 |
| Infantile spitting up | 0 | 1 (0.9) | 0.999 | 3 (2.4) | 4 (3.9) | 0.704 |
| Subjects with ≥ one Serious AE | 2 (1.8) | 3 (2.6) | 0.999 | 3 (2.4) | 2 (1.9) | 0.999 |
| Subjects with ≥ one AE Leading to Formula Discontinuation | 6 (5.3) | 8 (6.9) | 0.784 | 8 (6.4) | 5 (4.9) | 0.776 |
| Subjects with ≥ one Adverse Event of Special Interest (AESI) | 26 (22.8) | 31 (26.7) | 0.543 | 33 (26.4) | 25 (24.3) | 0.761 |
| Gastrointestinal Disorders | | | | | | |
| Gastroesophageal reflux disease | 8 (7.0) | 13 (11.2) | 0.361 | 10 (8.0) | 9 (8.7) | 0.999 |
| Vomiting | 5 (4.4) | 4 (3.4) | 0.747 | 6 (4.8) | 2 (1.9) | 0.299 |
| Constipation | 2 (1.8) | 6 (5.2) | 0.281 | 2 (1.6) | 2 (1.9) | 0.999 |
| Infantile spitting up | 0 | 2 (1.7) | 0.498 | 4 (3.2) | 6 (5.8) | 0.353 |
| Skin and Subcutaneous Disorders | | | | | | |
| Eczema/ infantile eczema | 4 (3.5) | 4 (3.4) | 0.999 | 4 (3.2) | 1 (1.0) | 0.381 |
| Dermatitis diaper | 7 (6.1) | 0 | 0.007* | 8 (6.4) | 3 (2.9) | 0.353 |

Abbreviations: AE, adverse event; AESI, adverse event of special interest.

[1]Adverse events were coded using the Medical Dictionary for Regulatory Activities (MedDRA) versions 20.0 and 23.0 for Parts 1 and 2, respectively. Only adverse events that occurred in ≥ 5% of the population in either Part 1 or 2 are shown, all other adverse events are available in the **Supplemental Data** in S1 Data.

[2]*Analyzed via two-sided Fisher's exact test, asterisk indicates significant difference from the Brand group at α = 0.05.

The total incidence of AEs was similar across all groups. Compared to the Brand formula group, the BBN-001 group reported fewer cases of diaper dermatitis, while the BBN-102 group had lower incidences of moderate/severe AEs or formula-related AEs. Of AEs related to formula, the BBN-102 exhibited lower incidences of flatulence, irritability, gastro-esophageal reflux disease, and vomiting, though no specific AE was statistically lower than the Brand group. Generally, the rates of AEs in this study are low. For example, there were 3/116 (2.6%) and 2/103 (1.9%) subjects with at least one serious AE in the BBN-001 and BBN-102 formulas, respectively. In comparison, Qian et al. [35] meta-analyzed rates of serious AEs among infants fed formulas containing probiotics, prebiotics, or synbiotics and found rates of serious AEs at 57/748 (7.6%). Qian et al.'s meta-analysis [35] found higher rates of non-serious AEs related to diarrhea (215/1297, 16.6%) and vomiting (34/843, 4.0%) in experimental formulas than in our experimental formulas, as the present study found low rates of AEs related to diarrhea (0% in BBN-001 and BBN-102) and vomiting (BBN-001: 3/116, 2.6%; BBN-102:

**Part 1**　　　　　　　**Part 2**

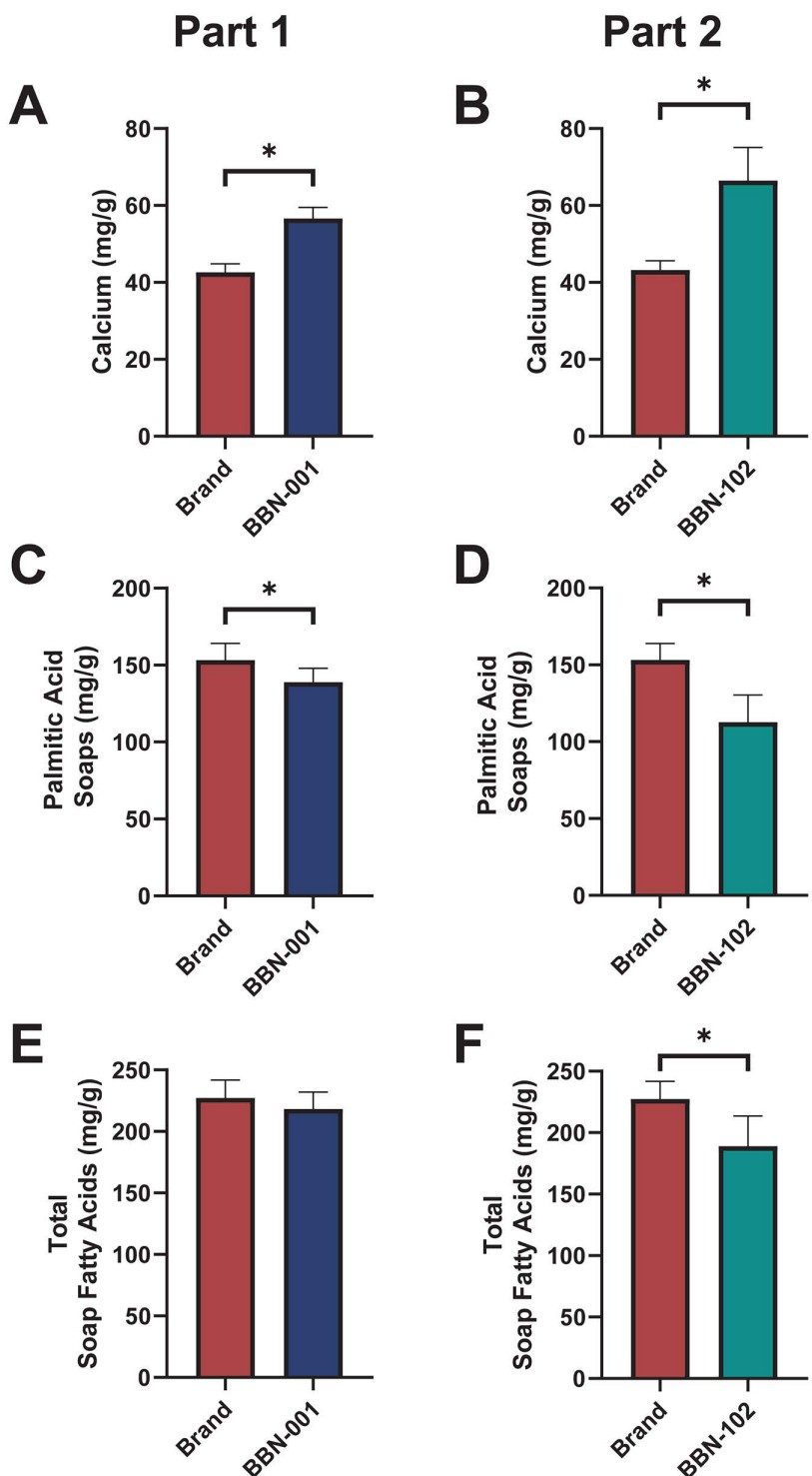

**Fig 5. Stool composition of the Safety population.** Data were analyzed on a dry weight basis. Data analyzed by two-sample t-test. *Indicates significance at $P < 0.05$. Values indicate the mean ± 95% confidence interval.

1/103, 1.0%) [35]. Overall Qian et al., found no differences between experimental and control formulas in rates of any type of AE [35].

Evaluation of GI tolerance showed significantly lower gassiness in infants fed either experimental formula compared to the Brand formula, while the BBN-102 group had lower incidence of fussiness as well. There were no significant differences between BBN-001 and Brand formula groups in terms of parent-reported ICQ scores for fussy/difficult, unpredictable, unadaptable, or dull sub-scores, while the BBN-102 group had lower ratings for the fussy/difficult subscale than the Brand formula group. It's unclear if these are related to the findings on stool characteristics.

Stool analysis revealed that infants fed the BBN-001 formula had significantly higher calcium concentrations and lower palmitic acid soaps concentrations at Visit 9 compared to those fed the Brand formula. However, there was no significant difference in the mean total soap fatty acid concentration between the groups. The BBN-102 group exhibited higher fecal calcium, and lower palmitic and total soap fatty acids than the Brand formula group. The present study's reductions in total soap fatty acids (9–38 mg/g) and palmitic acid soaps (14–40 mg/g), along with no change in stool consistency or frequency, are consistent (though more mild) with a meta-analysis of sn-2 palmitate–enriched formulas, which reported similar reductions (soap fatty acids: –34.7 [–50.8, –18.6] mg/g; palmitic acid soaps: –33.1 [–40.6, –25.6] mg/g) and no effects on stool consistency/frequency [32]. While inclusion of OPO in formulas has been cited to decrease the formation of insoluble fatty acid soaps and improve mineral absorption [5,36,37], theorized to lead to softer stools, the softening of stool is not supported by Zhang et al.'s meta-analysis [32]. Softer stool consistency in infants fed OPO at ≤ 2 mo of age are not maintained by 4 mo of age [32], the age at which they were measured in the present study. When compared to standard formulas, Zhang et al. found sn-2 palmitate–enriched formulas to reduce soap fatty acids (C12:0, C16:0, and total soap fatty acids). However, they found fewer differences compared to human milk-fed infants except higher C18:1 and C18:2 soap fatty acids [32], ultimately suggesting sn-2 enriched formulas shift the stool composition towards that of human milk-fed infants and away from those fed standard formulas. Of course, the current study cannot causally attribute the results to intake of sn-2 given the multi-component nature of the formulas.

It may appear contradictory that the present study found an increase in total fecal calcium, however this is different from fatty acids saponified with calcium (soap fatty acids), of which the present results are in-line with past trials [32]. Unfortunately, Zhang et al.'s meta-analysis [32] was unable to make conclusions on the effect of sn-2 palmitate–enriched formulas on calcium absorption due to the heterogeneity of the data and limited number of trials that measured calcium absorption. It's possible that the increases in fecal calcium simultaneous with decreases in palmitic acid soaps and total soap fatty acids may be explained by the inclusion of OPO in the experimental formulas and higher levels of prebiotics in the Brand formulas. While calcium content was the same between all formulas (549 mg/L), the Brand formula contained 4 g/L of prebiotics compared to 0.9 and 2.3 g/L in the BBN-001 and BBN-102 formulas, respectively (though each group consumed different amounts of GOS, PDX, and FOS). Both experimental formulas contained 4.9 g OPO compared to none in the Brand formula. A previous trial found that infants fed Sn-2 + oligofructose (3 g/L) had lesser stool calcium (28.5 mg/g) than controls or sn-2 fed infants (38–39.4 mg/g), respectively, while either sn-2 formula reduced palmitate soaps [38]. This aligns with a previous trial demonstrating that sn-2 enrichment of formula had minimal impact on total fecal calcium [39]. Such studies indicate that the prebiotic, and not sn-2, is related to changes in total fecal calcium. We speculate that, in the present study, the increase in total fecal calcium might reflect reductions in prebiotic intake in experimental groups, while the decrease in soap fatty acids is related to consumption of OPO.

### 4.1 Limitations

Given the non-inferiority design of the trial and methodological differences, comparisons between the BBN-001 and BBN-102 groups were not made, precluding statistical inference of the similarity between experimental formulas. While intentionally designed as a non-inferiority study comparing experimental formulas, future research could benefit from inclusion of a breastfed reference group to contextualize the results found. Given the multi-component nature of the formulas, the

study lacks the ability to make causal inferences about which components were responsible for the observed effects. Further, there was slightly unequal randomization in Part 2 leading to some differences in subject characteristics at enrollment. Dropout rates of 26–32.5% were accounted for by estimating for 25% attrition in the statistical analysis plan based upon the rate of dropout in previous studies of similar designs [14,40,41]. Trials on other experimental formulas have also shown dropout rates in the ranges of 23–35% [7,14,15,22], suggesting the dropout rates in this study were typical. The study was limited in its composition and assessment of geographical, socioeconomic, ethnic, maternal/parental, and genetic factors. Additionally, infants were exclusively fed cow's milk-based infant formula during the trial. For these reasons, results should be interpreted with care, as they are largely representative of infants from the general population exclusively fed cow's milk-based infant formula, as opposed to mixed-fed infants or those consuming other types of formula (e.g., soy-based, amino acid-based, or hypoallergenic formulas).

## 5. Conclusions

The experimental formulas containing unique ingredients (OPO, α-lactalbumin, lactoferrin, OPN, PDX, GOS, FOS, lutein, and β-carotene) were found to be non-inferior to a Brand formula in terms of weight, length, and head circumference gain velocities in a randomized, controlled, double-blind, multicenter, non-inferiority design safety study on healthy, term infants. Infants fed the experimental formulas had similar or fewer adverse events compared to those fed the Brand formula. Although there were differences in fecal calcium excretion and fatty acid concentrations, the experimental formulas were well-tolerated and deemed non-inferior to the Brand formula. However, infants fed the experimental formulas demonstrated similar bowel movements and consistency as the Brand formulas, with similar or fewer parent reports of gassiness and fussiness. Based on these data, we concluded that the experimental formulas were well-tolerated and non-inferior to the commercially available formulas.

## Supporting information

**S1 File. Supplemental Material.**
(DOCX)

**S1 Data. Supplemental Data.**
(XLSX)

**S1 Protocol. Trial Protocol.**
(PDF)

**S1 Checklist. CONSORT checklist.**
(DOCX)

## Acknowledgments

The authors thank Kevin L. Keim, Paidion Research, Inc., for medical monitoring and Ryan McBride (RM), Instat Clinical Research, for statistical analysis.

## Author contributions

**Conceptualization:** James McGrath.

**Funding acquisition:** Anne S. Kvistgaard, James McGrath.

**Project administration:** James McGrath, David K. Glover.

**Supervision:** James McGrath, David K. Glover.

**Visualization:** Stephen A. Fleming.

**Writing – original draft:** Stephen A. Fleming, Stefanie Flunkert.

**Writing – review & editing:** Stephen A. Fleming, Stefanie Flunkert, Anne S. Kvistgaard, James McGrath, David K. Glover.

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
