## [Decision Letter · Decision Letter 0]

15 Sep 2025

New Infant Formulas for Healthy Term Infants: A Randomized, Controlled, Double-Blind, Multicenter, Non-inferiority Design Safety Study

PLOS ONE

Dear Dr. Fleming,

Thank you for submitting your manuscript to PLOS ONE. After careful consideration, we feel that it has merit but does not fully meet PLOS ONE’s publication criteria as it currently stands. Therefore, we invite you to submit a revised version of the manuscript that addresses the points raised during the review process.

https://journals.plos.org/plosone/s/submission-guidelines#loc-laboratory-protocols . Additionally, PLOS ONE offers an option for publishing peer-reviewed Lab Protocol articles, which describe protocols hosted on protocols.io. Read more information on sharing protocols at https://plos.org/protocols?utm_medium=editorial-email&utm_source=authorletters&utm_campaign=protocols .

We look forward to receiving your revised manuscript.

Kind regards,

Afagh Hassanzadeh Rad

Academic Editor

PLOS ONE

Journal Requirements:

3. Thank you for uploading your study's underlying data set. Unfortunately, the repository you have noted in your Data Availability statement does not qualify as an acceptable data repository according to PLOS's standards.

Reviewer's Responses to Questions

**Comments to the Author**

1. Is the manuscript technically sound, and do the data support the conclusions?

Reviewer #1: Yes

Reviewer #2: Yes

Reviewer #3: Partly

2. Has the statistical analysis been performed appropriately and rigorously?

Reviewer #1: I Don't Know

Reviewer #2: Yes

Reviewer #3: Yes

3. Have the authors made all data underlying the findings in their manuscript fully available?

Reviewer #1: No

Reviewer #2: Yes

Reviewer #3: Yes

4. Is the manuscript presented in an intelligible fashion and written in standard English?

Reviewer #1: Yes

Reviewer #2: Yes

Reviewer #3: Yes

Reviewer #1: subject

The purpose of the study was well stated and was well designed.

Suggested:

The study covers a 120day study and the long -term effects of this formula can provide more information that can be mentioned in the restrictions section,

Not given the geographical or ethnic diversity of infants and its effect on information results

Reviewer #2: The complexity of a two-part trial requires careful explanation (which the authors have done, but in a clearer way is needed).

- A key limitation of this study is that both experimental formulas differed from the control on multiple components simultaneously. Because several compositional changes were tested together, it is not possible to determine which specific ingredient(s) were responsible for the observed improvements in tolerance (reduced gassiness/fussiness) or stool fatty acid profiles.

- Another limitation is that genetic and parental influences on infant growth were not evaluated. Since anthropometric factors are strongly influenced by heredity, some observed differences in growth patterns may reflect underlying genetic variation rather than formula composition. Although baseline size was adjusted in statistical models, without genetic or parental growth data, residual confounding remains possible.

- The experimental formulas differ in multiple components, making attribution of effects difficult. At the same time, the relatively small control groups (especially in Part 2 due to 8:1 randomization) reduce statistical precision. Together, these design features limit confidence in interpreting the observed differences.

- The study considered breastfeeding only in terms of duration prior to enrollment, but did not account for other maternal or contextual influences (such as maternal nutrition, health status, or socioeconomic conditions) that are known to affect infant growth through breast milk composition and feeding practices. It limits the ability to generalize findings to mixed-fed populations and to compare outcomes meaningfully against the full range of breastfeeding influences.

- While this study demonstrates safety and non-inferiority of the new formulas compared to standard formula, future research should include a concurrent breastfed reference group. Breastfeeding is the gold standard for infant nutrition, and having an exclusively breastfed control would allow for more meaningful interpretation of growth and tolerance outcomes in the context of natural feeding. Such a comparison would also help clarify whether any observed differences between formulas approach, match, or diverge from the growth patterns of breastfed infants.

Reviewer #3: Dear editor in chief,

This manuscript can be accepted. It was a nice and comprehensive article on formulas for infants. The results can shed light on this issue and can be used to be evaluated in further investigations.

**Do you want your identity to be public for this peer review?** For information about this choice, including consent withdrawal, please see our Privacy Policy

Reviewer #1: No

Reviewer #2: **Yes: ** Dr. Saeid Sadat mansouri

Reviewer #3: No

---

## [Decision Letter · Decision Letter 1]

30 Oct 2025

New infant formulas for healthy term infants: A randomized, controlled, double-blind, multicenter, non-inferiority design safety study

PONE-D-25-44390R1

Dear Dr. Fleming,

We’re pleased to inform you that your manuscript has been judged scientifically suitable for publication and will be formally accepted for publication once it meets all outstanding technical requirements.

Kind regards,

Afagh Hassanzadeh Rad

Academic Editor

PLOS ONE

Additional Editor Comments (optional):

Reviewers' comments:

Reviewer's Responses to Questions

**Comments to the Author**

Reviewer #1: All comments have been addressed

Reviewer #2: All comments have been addressed

Reviewer #4: (No Response)

2. Is the manuscript technically sound, and do the data support the conclusions?

Reviewer #1: Partly

Reviewer #2: Yes

Reviewer #4: Partly

3. Has the statistical analysis been performed appropriately and rigorously?

Reviewer #1: I Don't Know

Reviewer #2: Yes

Reviewer #4: No

4. Have the authors made all data underlying the findings in their manuscript fully available?

Reviewer #1: Yes

Reviewer #2: Yes

Reviewer #4: Yes

5. Is the manuscript presented in an intelligible fashion and written in standard English?

Reviewer #1: Yes

Reviewer #2: Yes

Reviewer #4: Yes

Reviewer #1: Dear editor in chief,

I have checked the revision and it is accepted and you can publish it

Thank you for your invitation

Reviewer #2: (No Response)

Reviewer #4: In addition to study design concerns raised during an earlier review, there are issues with the sample size and power estimates (Section 2.8.2).

For Part 1, the text reports a sample size of 90 per formula group (180 total) is needed to give 80% power if the standard deviation is assumed to be 5.6 g/d, but with a larger standard deviation of 6.0 g/d (the observed standard deviation at the time of interim analysis) 90 subjects per group would provide 90% power. I was not able to replicate the calculations reported, but in any case, given a larger standard deviation and the same sample size, the power would be lower not greater.

For Part 2, the control group is not truly a randomized control group, but rather the control group from Part 1 (conducted 3+ years earlier) supplemented with a small number of participants added for Part 2. This is likely largely responsible for the differences in subject characteristics between the Brand and BBN-102 infants (lines 318 – 321). This lack of true randomization for Part 2 invalidates the ‘independent, identically distributed’ statistical assumption underlying the statistical tests and thus calls into question the interpretation of the reported results.

Taken together with study design concerns raised by previous reviewer #2, the ability to draw meaningful conclusions from this report is limited.

**Do you want your identity to be public for this peer review?** For information about this choice, including consent withdrawal, please see our Privacy Policy

Reviewer #1: No

Reviewer #2: **Yes: ** Saeid Sadat Mansouri

Reviewer #4: No

---

## [Editor Report · Acceptance letter]

PONE-D-25-44390R1

PLOS ONE

Dear Dr. Fleming,

I'm pleased to inform you that your manuscript has been deemed suitable for publication in PLOS ONE. Congratulations! Your manuscript is now being handed over to our production team.

Kind regards,

on behalf of

Dr. Afagh Hassanzadeh Rad

Academic Editor

PLOS ONE